

# Comparative analysis of the complete plastid genomes of *Mangifera* species and gene transfer between plastid and mitochondrial genomes

Yingfeng Niu[1], Chengwen Gao[2] and Jin Liu[1]

[1] Yunnan Institute of Tropical Crops, Xishuangbanna, China
[2] The Affiliated Hospital of Qingdao University, Qingdao, China

## ABSTRACT

Mango is an important commercial fruit crop belonging to the genus Mangifera. In this study, we reported and compared four newly sequenced plastid genomes of the genus Mangifera, which showed high similarities in overall size (157,780–157,853 bp), genome structure, gene order, and gene content. Three mutation hotspots (*trnG-psbZ, psbD-trnT*, and *ycf4-cemA*) were identified as candidate DNA barcodes for Mangifera. These three DNA barcode candidate sequences have high species identification ability. We also identified 12 large fragments that were transferred from the plastid genome to the mitochondrial genome, and found that the similarity was more than 99%. The total size of the transferred fragment was 35,652 bp, accounting for 22.6% of the plastid genome. Fifteen intact chloroplast genes, four tRNAs and numerous partial genes and intergenic spacer regions were identified. There are many of these genes transferred from mitochondria to the chloroplast in other species genomes. Phylogenetic analysis based on whole plastid genome data provided a high support value, and the interspecies relationships within Mangifera were resolved well.

## INTRODUCTION

Mango is a tall, evergreen tree belonging to the genus *Mangifera* of the Anacardiaceae family. It is an important tropical fruit (*Iquebal et al., 2017*; *Lora & Hormaza, 2018*) that originates in tropical and subtropical regions in Southeast Asia (*Dutta et al., 2013*; *Sherman et al., 2015*). Owing to its wide range of cultivation (*Bajpai et al., 2016*), high nutrient value, pleasing appearance, and unique flavor (*Surapaneni et al., 2013*), it is widely loved by consumers and has the reputation of being known as the "King of Tropical Fruits" (*Khan, Ali & Khan, 2015*). Southeast Asian countries have a history of mango cultivation that spans thousands of years (*Ravishankar et al., 2013*). Mangoes were introduced to Africa, South America, and other continents hundreds of years ago, and several varieties suitable for local cultivation have been developed (*Mansour, Mekki & Hussein, 2014*; *Sennhenn et al., 2014*). There are 69 species of mango in the world that are mainly distributed in tropical and subtropical countries including India, Indonesia, the Malay Peninsula, Thailand, and

Corresponding authors
Chengwen Gao,
gaochengwen6@126.com
Jin Liu, liujin06@126.com

South China, of which, five species are grown in China, namely *M. indica, M. persiciformis, M. longipes, M. hiemalis,* and *M. sylvatica*; however, the varieties cultivated in production belong to *M. indica*. Phylogenetic analysis of *Mangifera* species has been a hot topic of research (*Nishiyama et al., 2006*; *Sankaran et al., 2018*), while the whole chloroplast genome sequences can provide more genetic information and higher species resolution ability than other molecular data. However, the chloroplast genomes of most *Mangifera* plants remain unknown (*Azim, Khan & Zhang, 2014*).

Chloroplasts are special organelles that are involved in photosynthesis and consist of layers of thylakoids. They have their own DNA and can split. The chloroplast genome is conserved and consists of four parts. Two inverted repeat (IR) regions separate the small copy region (SSC) and large copy region (LSC). Currently, with the rapid development of next-generation sequencing (NGS) technology, the entire chloroplast genome has been widely used for phylogenetic analysis. They can provide a large number of variable sites for phylogenetic analysis (*Gitzendanner et al., 2018*). Thus, the entire chloroplast genome shows the potential to resolve evolutionary relationships and produce highly resolved phylogenetic and genetic diversity, particularly in some complex taxa or at low taxonomic levels, which have unresolved relationships (*Hu et al., 2016*; *Huang et al., 2020*; *Xu et al., 2019*).

In this study, the chloroplast genomes of four *Mangifera* species were sequenced and compared with *M. Indica* and 21 Sapindales plastids. The objectives of this study were as follows: (1) to comparatively analyze the chloroplast genome structure of five species of *Mangifera;* (2) to identify highly divergent regions of the chloroplast genomes of *Mangifera*; (3) to determine the insertion of chloroplast genes into mitochondria; (4) to explore the evolutionary relationship between the genus, *Mangifera,* and Sapindales. Overall, this study would be helpful to further understand plastid evolution and phylogeny of the genus, *Mangifera*.

## MATERIALS AND METHODS

### Plant material, DNA extraction, and sequencing

Fresh leaves of four *Mangifera* species (*M. hiemalis*, *M. persiciformis*, *M. longipes*, and *M. sylvatica*) were collected from Xishuangbanna Tropical Flowers and Plants Garden, South Yunnan, China, and frozen in liquid nitrogen. Total genomic DNA was extracted from all samples according to CTAB method (*Li et al., 2013*). DNA quality was detected using 1% agarose gel electrophoresis and samples were stored at $-80\,°C$ until further use.

About 5–10 μg of total DNA were extracted from each of the *Mangifera* samples to construct a shotgun library with an average insertion size of 300 bp. Paired-end libraries were constructed with NEBNext® DNA Library Prep Master Mix Set for Illumina according to the manufacturer's recommendation. Illumina HiSeq 2500 system (Illumina, San Diego, CA, USA) was used to sequence DNA samples in the paired-end sequencing mode by Novogene Bioinformatics Technology Co. Ltd (Beijing, China), generating approximately 8.0 Gb of raw data per sample. The plastome depth of coverage was more than $2000\times$.

## Chloroplast genome assembly and annotation

The Trimmomatic v0.38 was used to filter raw sequencing data (*Bolger, Lohse & Usadel, 2014*), and the obtained clean data were de novo assembled using SPAdes v3.61 under different K-mer parameters (*Bankevich et al., 2012*). The scaffolds that were positively associated with chloroplasts were arranged on the reference chloroplast genome of *M. indica* (NC_035239). Paired-end reads were remapped to consensus assembly and multiple iterations were performed to fill in the gaps in the final consensus sequence using Geneious software v2020.0.4 (*Kearse et al., 2012*).

Chloroplast genome annotation was performed using GeSeq (https://chlorobox.mpimp-golm.mpg.de/geseq.html) to predict genes encoding proteins, transfer RNA (tRNA), and ribosomal RNA (rRNA), and was adjusted manually as needed (*Tillich et al., 2017*). We also manually examined the IR junctions of all *Mangifera* species. A circular diagram of the chloroplast genomes of *Mangifera* was subsequently drawn using OGDRAW v1.3.1 (*Greiner, Lehwark & Bock, 2019*).

## Genome comparative analysis and divergent hotspot identification

MAFFT v7.221 was used to align the chloroplast genome sequences of five *Mangifera* plants (*Katoh & Standley, 2013*). Next, DnaSP v6.12 was used to perform a sliding window analysis with the step size of 200 bp and window length of 600 bp, to detect the rapidly evolving molecular markers for performing phylogenetic analysis (*Librado & Rozas, 2009*).

## Identification of chloroplast gene insertion in mitochondria

First, we removed the BLAST hits of genes transferred between chloroplast and mitochondrial genomes by mapping the mitochondrial genome of *M. indica* (GenBank: CM021857) to the plastid genomes. Circos v0.69-9 (*Krzywinski et al., 2009*) software was used to map the mitochondrial and chloroplast genomes of the *Mangifera* species as well as gene-transfer fragments.

## Phylogenetic analysis

Phylogenetic analyses were performed for five *Mangifera* (4 species sequenced here) and 21 Sapindales species, using *Arabidopsis thaliana* as outgroups. MAFFT 7.221 (*Katoh & Standley, 2013*) was used to align the chloroplast genome sequences of Sapindales species. We used the following three methods to perform phylogenetic analyses of *Mangifera* species: Bayesian Inference (BI) with a GTR + I + G model using MrBayes v3.2 (*Ronquist et al., 2012*), the Markov chain Monte Carlo (MCMC) algorithm was run for 1 million generations and sampled every 100 generations. Maximum Likelihood (ML) using MEGA v7.0 with 1000 bootstrap replicates (*Kumar, Stecher & Tamura, 2016*), and Maximum Parsimony (MP) with a heuristic search in PAUP v4.0 with 1,000 random taxon stepwise addition sequences (*Swofford, 1993*). A 50% majority-rule consensus phylogeny was constructed using 1,000 bootstrap replications.

## RESULTS AND DISCUSSION

### Basic characteristics of the *Mangifera* chloroplast genomes

Raw data (approximately from $7.1 \times 10^9$ to $8.3 \times 10^9$ bp) were obtained from *M. hiemalis* (MN917208), *M. persiciformis* (MN917209), *M. longipes* (MN917210), and *M. sylvatica* (MN917211). The four newly sequenced *Mangifera* chloroplast genomes have been presented to the GenBank database.

Characteristics of four newly sequenced and one reported *Mangifera* chloroplast genomes were investigated. *Mangifera* chloroplast genome sequence sizes were 157,780~157,853 bp (Fig. 1), with the largest and smallest being those of *M. longipes* and *M. indica,* respectively. *Mangifera* chloroplast genomes are characterized by a typical four-part structure, two IR copies (26,354–26,379 bp) separating the LSC (86,673–86,726 bp) and SSC (18,347–18,369 bp) regions. In addition, the GC content of *Mangifera* genomes was similar, ranging from 37.88–37.89%. Five *Mangifera* chloroplast genomes contained 113 predicted functional genes, including 79 protein-coding genes, four ribosomal RNA (rRNA) genes, and 30 transfer RNA (tRNA) genes (Tables 1 and 2). Furthermore, 15 functional genes, including four protein-coding genes, four ribosomal RNA genes, and seven transfer RNA gene replicate in the IR regions of the chloroplast genome. The number, type, and order of genes were found to be very similar among the five *Mangifera* chloroplast genomes (*Jo et al., 2017*; *Rabah et al., 2017*; *Zhang et al., 2020*). The whole chloroplast genome sequences of four *Mangifera* species were submitted to GenBank with the accession numbers of MN917208 to MN917211.

The IR/SC connected regions were found nearly identical relative positions in the five *Mangifera* chloroplast genomes (Fig. 2). All LSC-IRb connections were found to be located within the *rps19* gene, resulting in a partial expansion of the IRb region to the *rps19* gene (80–104 bp). The IRb-SSC boundary was located in the *ndhF* gene, while the SSC-IRa boundary in the five chloroplast genomes was located in the *ycf1* gene.

### Comparative *Mangifera* chloroplast genomes and Divergence Hotspot Regions

Using the comparative sequence analysis of the five species of *Mangifera*, we found that the plastid genome was quite conservative in the five taxa, although there were a few regions with variations. In general, sequences are conserved in the coding region, and most of the detected variations are in the non-coding region. The results agree with previous reports that non-coding regions showed greater divergence than coding regions, this is possibly caused by coding regions affected by stronger selective pressure (*Li et al., 2018*). Consistent with similar studies involving other plants, the IR regions appear to be more conservative than the LSC and SSC regions (Fig. 1) (*Liang et al., 2019*; *Song et al., 2019*). A search for nucleotide substitutions identified 638 variable sites (0.40%) in the five chloroplast genomes, including 489 parsimony-informative sites (0.31%), this number is smaller than other genus species (*Gao et al., 2020*; *Nguyen et al., 2020*).

To identify hotspots of sequence divergence, the nucleotide diversity (Pi) values within the 600 bp window of the *Mangifera* chloroplast genomes were calculated (Fig. 3). We found that Pi values varied from 0–0.033, and the three hypervariable regions (Pi > 0.02)

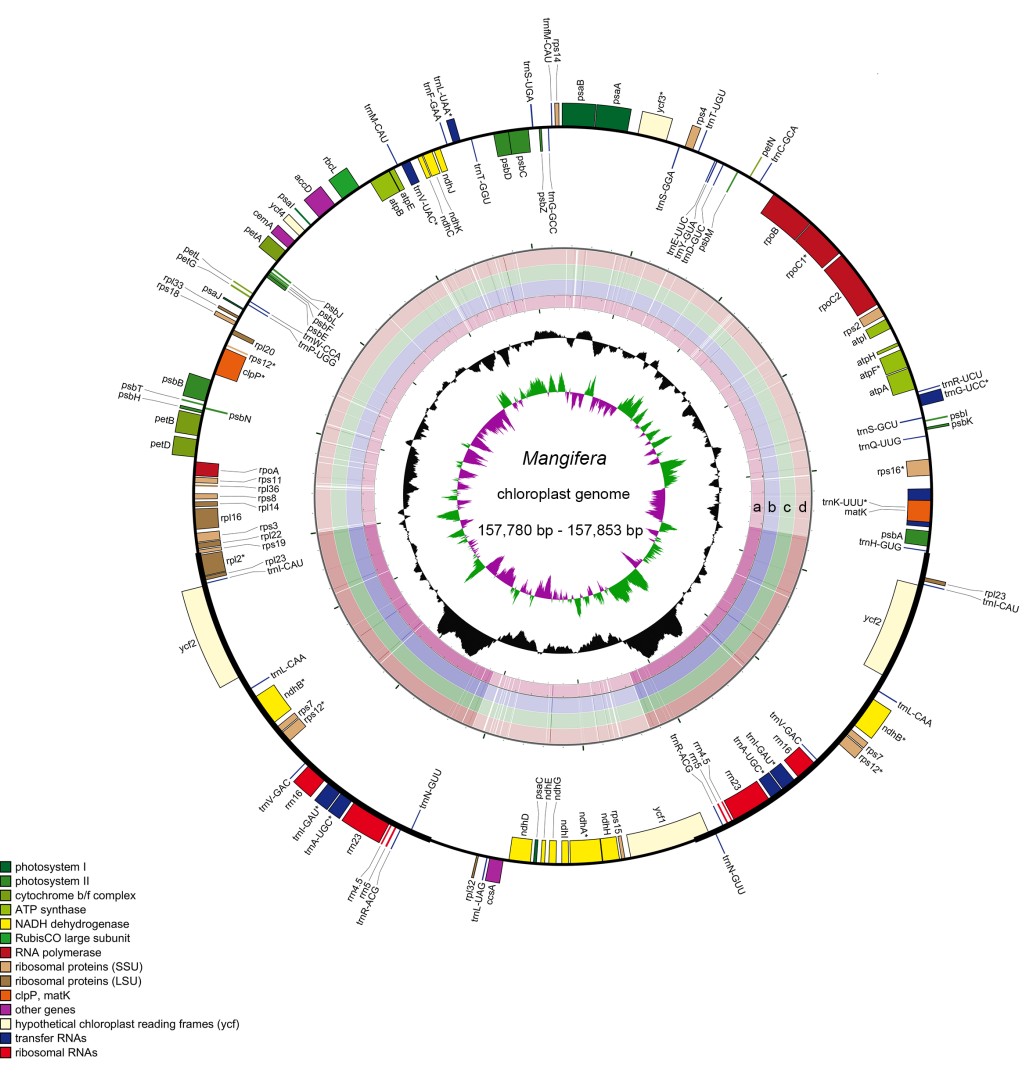

**Figure 1** **Sequence diagram of *Mangifera* chloroplast genomes.** Gene map of *Mangifera* chloroplast genomes, sequence alignment of *Mangifera* species chloroplast genome ((A) *M. Sylvatica*, (B) *M. hiemalis*, (C) *M. longipes,* (D) *M. persiciformis* with reference to *M. indica*), GC content, and GC skew from the outside to inside.

of the five *Mangifera* chloroplast genomes were *trnG-psbZ*, *psbD-trnT*, and *ycf4-cemA*. The *trnG-psbZ* region exhibited the highest variability (7.44%).

Here, we found an increase in the number of variable sites in the following three specific regions based on the results of pairwise plastid genomic alignment and SNP analysis: *trnG-psbZ*, *psbD-trnT*, and *ycf4-cemA*. Thus, *Mangifera* species may be detected using these regions as novel candidate fragments. Figure S1 presents the graphical representation of these results using the ML method. These three DNA barcode candidate sequences have high species identification ability. However, further experiments are required to support this *Mangifera* plastid sequence data.

**Table 1** Summary of chloroplast genome features of five *Mangifera* species.

| Genome feature | *M. indica* | *M. longipes* | *M. persiciformis* | *M. hiemalis* | *M. sylvatica* |
|---|---|---|---|---|---|
| Total size (bp) | 157,780 | 157,853 | 157,799 | 157,796 | 157,824 |
| LSC Length (bp) | 86,673 | 86,726 | 86,724 | 86,718 | 86,719 |
| SSC Length (bp) | 18,349 | 18,369 | 18,367 | 18,368 | 18,347 |
| IR Length (bp) | 26,379 | 26,379 | 26,354 | 26,355 | 26,379 |
| Total Genes | 113 | 113 | 113 | 113 | 113 |
| Protein coding Genes | 79 | 79 | 79 | 79 | 79 |
| Structure RNAs | 34 | 34 | 34 | 34 | 34 |
| GC Content (%) | 37.89% | 37.88% | 37.88% | 37.89% | 37.89% |
| GenBank Accessions | NC035239 | MN917210 | MN917209 | MN917208 | MN917211 |

**Table 2** Genes contained in *Mangifera* chloroplast genome.

| Category | Group of genes | Name of genes |
|---|---|---|
| Self replication | Ribosomal RNA genes | *rrn4.5, rrn5, rrn16, rrn23* |
| | Small subunit of ribosome | *rps2, rps3, rps4, rps7, rps8, rps11, rps12, rps14, rps15, rps16, rps18, rps19* |
| | Transfer RNA genes | *trnR-UCU, trnS-GCU, trnA-UGC, trnC-GCA, trnF-GAA, trnG-GCC, trnG-UCC, trnD-GUC, trnE-UUC, trnH-GUG, trnN-GUU, trnP-UGG, trnQ-UUG, trnR-ACG, trnI-GAU, trnY-GUA, trnK-UUU, trnL-CAA, trnL-UAA, trnI-CAU, trnV-GAC, trnV-UAC, trnW-CCA, trnL-UAG, trnfM-CAU, trnM-CAU, trnS-GGA, trnS-UGA, trnT-GGU, trnT-UGU* |
| | DNA dependent RNA polymerase | *rpoA, rpoB, rpoC1, rpoC2* |
| | Large subunit of ribosome | *rpl2, rpl14, rpl16, rpl20, rpl22, rpl23, rpl32, rpl33, rpl36* |
| Photosynthesis | Subunits of photosystem I | *psaA, psaB, psaC, psaI, psaJ, ycf3, ycf4* |
| | Subunits of NADH-dehydrogenase | *ndhA, ndhB, ndhC, ndhD, ndhE, ndhF, ndhG, ndhH, ndhI, ndhJ, ndhK* |
| | Subunits of ATP synthase | *atpA, atpB, atpE, atpF, atpH, atpI* |
| | Subunits of photosystem II | *psbA, psbB, psbC, psbD, psbE, psbF, psbH, psbI, psbJ, psbK, psbL, psbM, psbN, psbT, psbZ* |
| | Subunits of cytochrome complex | *petA, petB, petD, petG, petL, petN* |
| | Protease | *clpP* |
| Other genes | Maturase | *matK* |
| | Acetyl-CoA-carboxylase c-type Cytochrome synthesis gene | *ccsA* |
| | Large subunit of rubisco | *rbcL* |
| | Envelop membrane protein | *cemA* |
| | Subunit of Acetyl-CoA-carboxylase | *accD* |
| | Hypothetical chloroplast | *ycf1, ycf2, ycf15* |

## Characterization of gene transfer of *Mangifera* chloroplast genome to mitochondrial genome

The mitochondrial genome of *M. indica* was obtained from GenBank and was 87,1458 bp in size, approximately 5.5 times that of the chloroplast genome consisting of 94 functional genes. We identified 12 large chloroplast genome fragments in the mitochondrial genome,

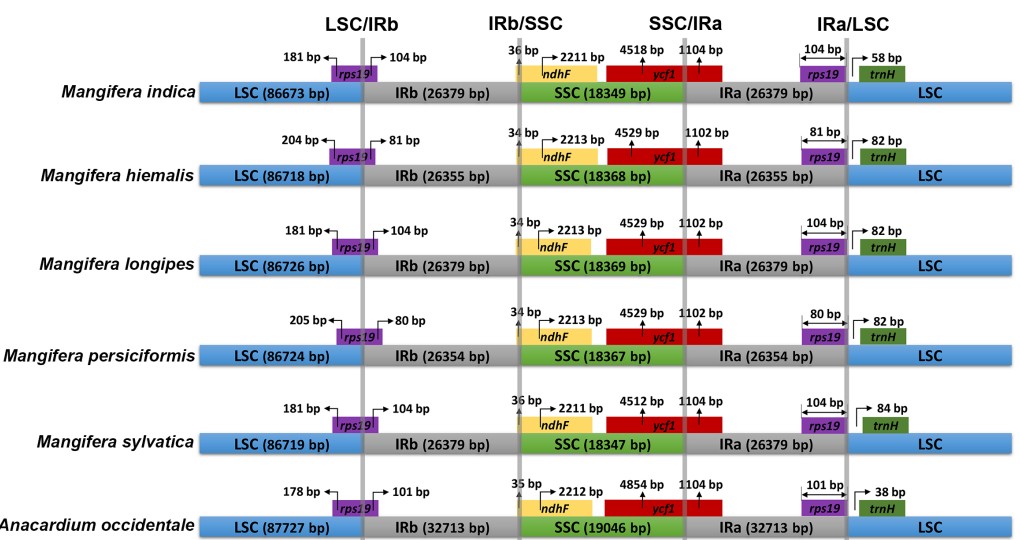

**Figure 2** **Comparison of inverted repeat (IR) boundary among *Mangifera* species, where genes and gene fragments across IRa/b junctions are represented in color boxes above the horizontal line.** Genes and IR segments are not mapped to scale.

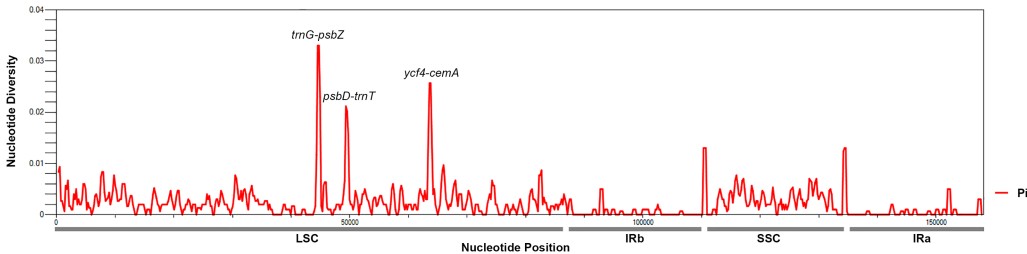

**Figure 3** ***Mangifera* Chloroplast genomes sliding window analysis (window length: 600 bp; step size: 200 bp).** *X*-axis, Position of a window; *Y*-axis, Genetic diversity per window.

including genes and intergenomic regions. These fragments ranged from 1,522–5,400 bp and the sequences were over 99% consistent. The total length of these fragments was 35,652 bp, accounting for 22.6% of the chloroplast genome (Fig. 4 and Table S1). Fifteen intact chloroplast genes (*rps19, rpl2, rpl23, petN, rbcL, accD, psbJ, psbL, psbF, psbE, petL, petG, psaA, atpA, cemA* ), four tRNAs (*trnI-CAU, trnC-GCA, trnW-CCA, trnP-UGG*) and numerous partial genes and intergenic spacer regions were identified. There are many of these genes transferred from mitochondria to the chloroplast in other species genomes, such as *rps12, rpl23, rbcL, petL, petG, trnW-CCA* and *trnP-UGG* (*Gao et al., 2020*; *Gui et al., 2016*).

Intracellular gene transfer exists between different genomes, including those of the chloroplasts, mitochondria, and nuclei (*Nguyen et al., 2020*; *Timmis et al., 2004*). Research shows that the frequency of nuclear DNA transfer from organelles in angiosperms is very high (*Hazkani-Covo, Zeller & Martin, 2010*; *Park et al., 2014*; *Smith, 2011*). Gene transfer
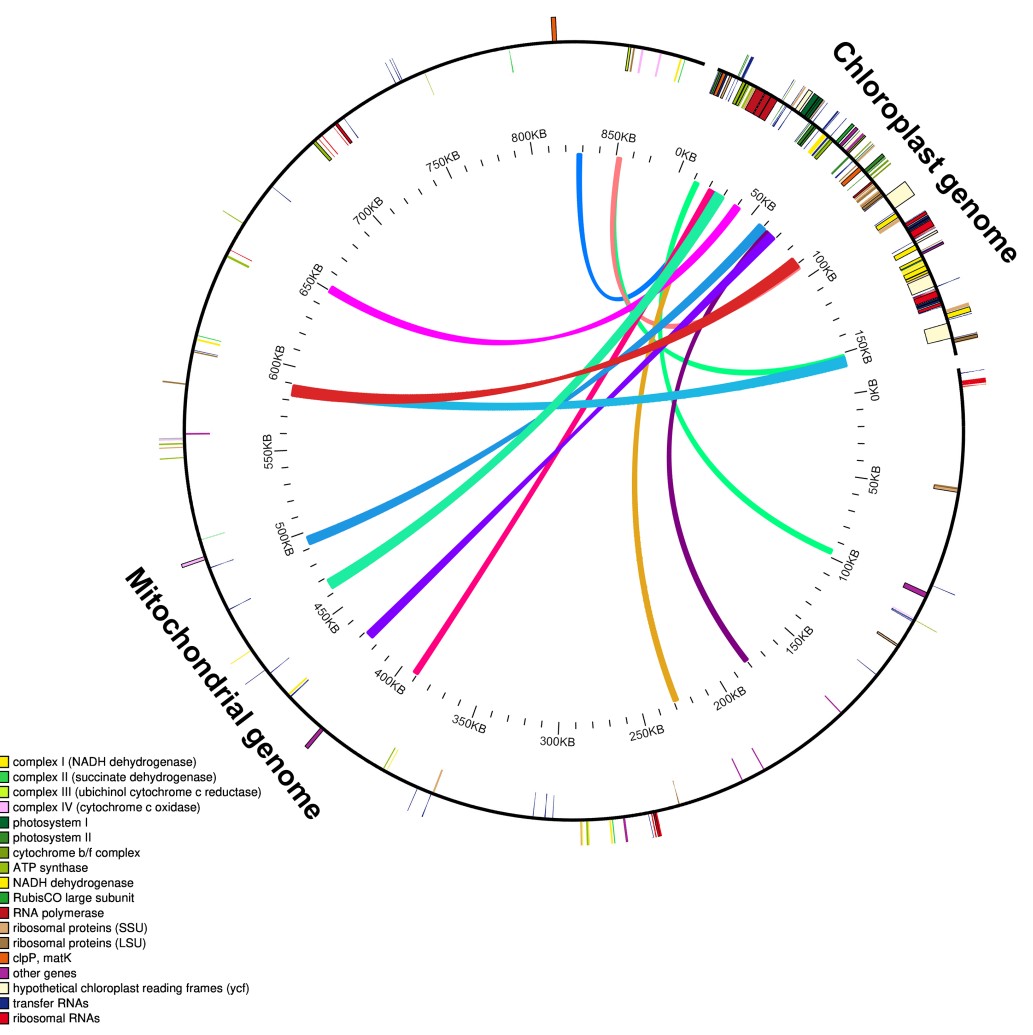

complex I (NADH dehydrogenase)
complex II (succinate dehydrogenase)
complex III (ubichinol cytochrome c reductase)
complex IV (cytochrome c oxidase)
photosystem I
photosystem II
cytochrome b/f complex
ATP synthase
NADH dehydrogenase
RubisCO large subunit
RNA polymerase
ribosomal proteins (SSU)
ribosomal proteins (LSU)
clpP, matK
other genes
hypothetical chloroplast reading frames (ycf)
transfer RNAs
ribosomal RNAs

**Figure 4**   **Schematic diagram of gene transfer between chloroplast and mitochondria in *Mangifera* species.**   Colored lines within the circle show where the chloroplast genome is inserted into the mitochondrial genome. Genes within a circle are transcribed clockwise, while those outside the circle are transcribed counterclockwise.

from chloroplast to mitochondrial genomes is a common phenomenon during long-term evolution (*Gui et al., 2016*; *Nguyen et al., 2020*). Due to high sequence identity between the transferred chloroplast genome fragments in the mitochondrial and original chloroplast genomes, gene transfer can lead to assembly errors in these genomes.

## Phylogenetic relationship of chloroplast genomes

In this study, the chloroplast genome was used for infer the phylogenetic location of *Mangifera* in Sapindales (Fig. 5) and performed a phylogenetic analysis of the chloroplast genome using three different methods, namely, ML, MP, and BI. BI and ML analyses revealed almost the same topology, and most branches had very high support (Fig. S2). However, MP trees differed slightly from BI and ML trees in some taxa (Fig. S3). Despite

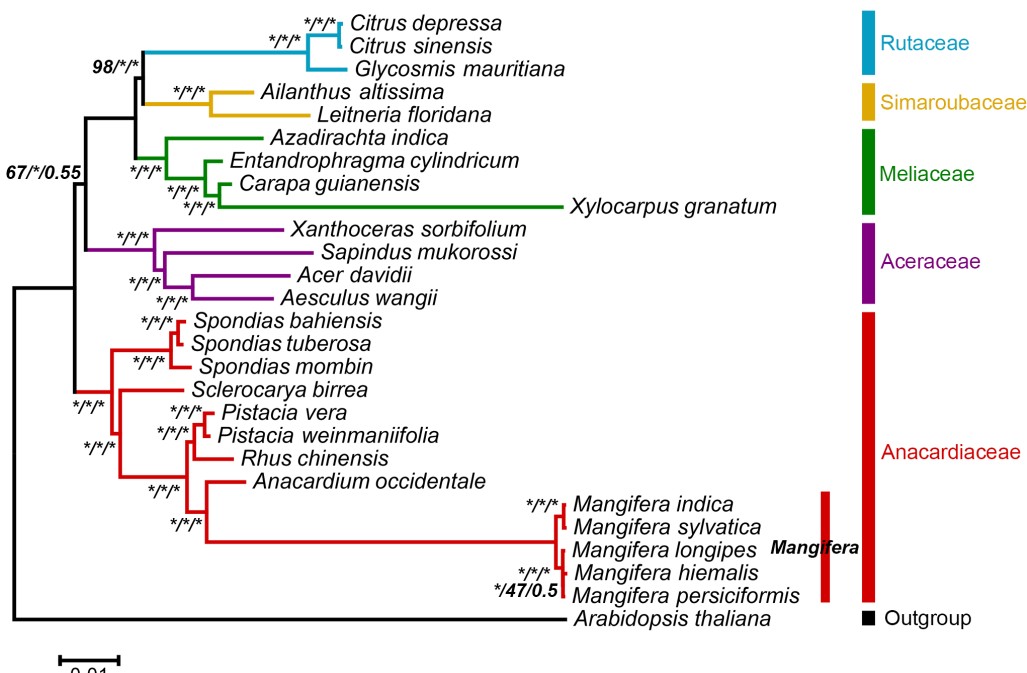

**Figure 5  ML phylogenetic tree of five *Mangifera* species with 21 related species in the Sapindales based on whole chloroplast genome sequence.** Numbers related to the branches are ML bootstrap value, MP bootstrap value, and Bayesian posterior probability, respectively. Asterisk denotes 100% bootstrap support or 1.0 posterior probability.

differences between these three approaches, the relationships between most groups were well resolved and highly supported, suggesting that the use of chloroplast genome data does significantly improve the resolution of phylogenetic analysis. Previous studies have revealed the genetic relationship of *Mangifera* through morphological, nuclear, amplified fragment length polymorphism, ribosomal internal transcribed spacer (ITS), and partial chloroplast gene analysis (*Eiadthong et al., 2000*; *Nishiyama et al., 2006*; *Sankaran et al., 2018*; *Yonemori et al., 2002*). The whole chloroplast genome sequence-based phylogenetic tree was built to explore the evolutionary similarities/differences between *Mangifera* species and between genera in the Sapindales. Phylogenetic analysis based on complete genome sequences, rather than a few genes, has been carried out in a large number of higher plant species, significantly improving the resolution of phylogenetic analysis (*Zhai et al., 2019*).

## CONCLUSIONS

In this study, the chloroplast genomes of four *Mangifera* species were sequenced and compared. It was found that the size, structure, and gene content of the *Mangifera* chloroplast genomes were conserved. Comparative analysis showed a low degree of sequence variation. We identified 13 large fragments that were transferred from the chloroplast genome to the mitochondrial genome. In addition, we identified three mutation hotspots as DNA barcodes for the identification of *Mangifera* species. These

complete chloroplast genome sequences and highly variable markers provide sufficient genetic information for the phylogenetic reconstruction and species identification of the genus *Mangifera*.

## ACKNOWLEDGEMENTS

We are grateful to thank Zhangguang Ni for the collection of experiment material.

### Funding

This work was supported by the Youth Talent Growth Fund of YITC (QNCZ2020-3), Technology Innovation Talents Project of Yunnan Province (2018HB086), and Sci-tech Innovation System Construction for Tropical Crops Grant of Yunnan Province (No. RF2021). The funders had no role in study design, data collection and analysis, decision to publish, or preparation of the manuscript.

### Grant Disclosures

The following grant information was disclosed by the authors:
The Youth Talent Growth Fund of YITC: QNCZ2020-3.
Technology Innovation Talents Project of Yunnan Province: 2018HB086.
Sci-tech Innovation System Construction for Tropical Crops Grant of Yunnan Province: No. RF2021.

### Competing Interests

The authors declare there are no competing interests.

### Author Contributions

- Yingfeng Niu conceived and designed the experiments, performed the experiments, authored or reviewed drafts of the paper, and approved the final draft.
- Chengwen Gao conceived and designed the experiments, performed the experiments, analyzed the data, prepared figures and/or tables, authored or reviewed drafts of the paper, and approved the final draft.
- Jin Liu conceived and designed the experiments, authored or reviewed drafts of the paper, and approved the final draft.

### DNA Deposition

The following information was supplied regarding the deposition of DNA sequences:
Data are available at GenBank: MN917208–MN917211.

### Data Availability

Data are available at NCBI: PRJNA655379.

## Supplemental Information

Supplemental information for this article can be found online at http://dx.doi.org/10.7717/peerj.10774#supplemental-information.

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
