# Peer review of "Comparative analysis of the complete plastid genomes of Mangifera species and gene transfer between plastid and mitochondrial genomes"

_PeerJ, doi:10.7717/peerj.10774_

## Round 0.1 · original submission · Major Revisions

Please take into consideration the reviewer’s comments and provide back a point-by-point rebuttal letter addressing those concerns.

In particular, evolutive and comparative aspects toward other plants, and the interplay between mitochondrial ad chloroplast genomes is key to emphasize the relevance of your study. The metrics of the phylogenetic analysis are also key to validate the inferences and conclusions.

Also, emphasize the availability of the chloroplast annotated genome in GenBank.

Reviewer 1 ·

Basic reporting

The manuscript deals with the importance of chloroplasts and mitochondria genomes to identify mango species from the genus Mangifera spp. The genomes from five mango species were described and compared. Some specific DNA fragments were identified as transferred from the chloroplast to mitochondrial genomes. The whole chloroplast genome phylogenetic analysis is suggested as a powerful tool to solve evolutionary relationships and discriminate among mango species.

Comments and suggestions:
The authors must review and correct the English language of the whole manuscript as it contains some grammatical, syntax, and spelling errors.
It is highly recommended to use passive voice through the manuscript; by using passive voice, readers will focus on the action and the results of the study. It is not really important for a reader to know who performed any action for the development of this study.
Some corrections:
-The excess of commas should be corrected.
-Lanes 33. No comma after “flavor”
-Lane 65. Published by Li et al. (2013)
-Lane 70. The illumina HiSeq 2500 system (XXXX) was used to sequence DNA samples in the paired-end…
-Lane 74. The Trimmomatic 0.38 tool was used
-Lane 93. Circos… software/ web tool?

Abstract.
-Lane´s 25-26 sentence gives an incomplete idea. A higher resolution for Magifera species identification?

Experimental design

Methods and Material.
-Basic and essential information is missing about materials and method.
-First paragraph. Authors should include how many, and the scientific names of those Mangifera species that were sampled.
- The method used for DNA isolation should be mentioned in the methods section.
- The authors should mention and describe the kit or reagents used to construct libraries.
- In the methods section, software or web tools are not clearly distinguished. The software may include the complete name and version; web tools may include electronic address.
- Phylogenetic analysis. Authors should describe better the analysis including:
¿how many other available chloroplast genomes were used to compare their Mangifera data?; ¿how, and why those sequences were selected?; ¿how authors evaluated the reliability of the tree topologies?; ¿Which were the optimized parameters for each selected method?

Validity of the findings

Results and Discussion
- Lane 106. The approximated data numbers indicate as units ¿nucleotides? Or ¿sequences? It should be mentioned.
- This section is a general results description, but the in-depth analysis of what these results mean should be improved.
- Why were Sapindales selected for comparison purposes?
- Why variations or differences among sequences are in the non-coding region of the chloroplast genomes? What does this mean?
- The identified hypervariable regions were analyzed in deep? What about any region that may function as a molecular marker allowing to distinguish among Mangifera species?
- Figure 4. The legend colors should be found in the little colored bars from the out/ in sides of the circle, however. the figure quality does not allow to clearly distinguish each color, so it makes difficult to understand the genes identity and location.
- Besides considering the assembly errors that mitochondrial genes transfer may produce in the chloroplast genome sequencing, authors should analyze in deep what do these transfers mean or promote into the chloroplast genome? ¿Are always the same genes transferred from mitochondria to the chloroplast in other species genomes? ¿Any of these transferred genes could be a specific marker from one Mangifera species?
The results from four main objectives of the study were obtained and described, but not explained.

Additional comments

In this manuscript, several original data were obtained. Some of them deserve to be better analyzed and discussed. The main differences among the sequenced genomes should be explained, and those differences with other genus species should also be considered in the discussion section.

Reviewer 2 ·

Basic reporting

The context is poorly provided in the Introduction. The first and second paragraph lack a clear connection. The knowledge gap is not described in the introduction.
The results need to be better described, especially the contrasting results.

Experimental design

- Provide more details about how these species were identified. Are they part of a botanical collection? Who identified these species?
- Please provide the names of kits used for library preparation. 5-10ug of DNA is a high amount of DNA. Were all of this amount used for library preps?
- The methods used for phylogenetic inference need more detail, for ex. was the substitution model choosed based on what?
- The phylogenetic inference was performed using species from 5 families. Theses species shows any structural rearrangements? how was the sequence alignment done?

Validity of the findings

- The sequencing details, as the plastome depth of coverage needs to be described.
- The phylogenetic inference methodology needs to be revised and the results and discussion on this topic rewritten.

Additional comments

- Please use "plastid genome" or "plastome" when referring to the DNA content of chloroplasts.
- Line 65. Revise Li et al. citation.
- Line 106. Provide the unit of measure for raw data.
- Line 108-109. "GenBank database"
- Lines 115-121. Please describe in detail the gene content variation.
- Lines 141-142. The variable sites were identified using only Mangifera plastomes. The statement that this region could be "ideal molecular markers to distinguish Mangifera from other genera" has no support.
Line. 155. Similarity or identity?
Line 159. Adjust, infer of confirm?

Reviewer 3 ·

Basic reporting

This work is interesting in terms of chloroplast genomes analysis between Mangifera species. Overall, the results are novel and presented in a logical order. There are a few points that have to be addressed by the authors towards improvement of the manuscript.
- Abstract should be rewritten to set a clear scope of the study, and not just report numbers and sizes.
- Authors should give info concerning the different Mangifera species used for chloroplast genome analysis.
Why did they select these species? Is there any special physiological or morphological characteristic or relation to cultivated mango? Authors should incorporate photographs from these plants describing their phenotype and/or diversity. There is a lot of space in Suppl. Figs.
- It would be beneficial to add plastome variation by region between the Mangifera species. SNPs and INDELs across coding gene regions, intergenic regions and introns could be described. Please add a graph about all the above.
- Since authors discovered possible “Divergence Hotspot Regions to be ideal molecular markers to distinguish Mangifera from other genera”, these markers should be experimentally verified and presented, otherwise please delete it.

Experimental design

Please see above.

Validity of the findings

Please see above.

Additional comments

Please see above.

---

## Round 0.2 · Minor Revisions

You are missing a citation for:

Azim, M.K., Khan, I.A. & Zhang, Y. Characterization of mango (Mangifera indica L.) transcriptome and chloroplast genome. Plant Mol Biol 85, 193–208 (2014). https://doi.org/10.1007/s11103-014-0179-8";

Reviewer 1 ·

Basic reporting

After the second revision, no fails were found!

Experimental design

After the second revision, no fails were found!

Validity of the findings

After the second revision, no fails were found!

Additional comments

Manuscript ID:
PeerJ-51712
Title: Comparative analysis of the complete chloroplast genomes of Mangifera species and gene transfer between chloroplast and mitochondrial genomes

The manuscript describes, compares, and discusses the chloroplast´s genome from various mango species. After revision, most corrections were done, and writing was improved. The previously missing and confusing data are precise now.

Minor comments:
Table 2. Column 1. Photosynthesis (P capitalized). Column 2. Cytochrome Synthesis ( capitalized and an “e” is missing).

---

## Round 0.3 · accepted · Accept

Thanks for addressing all the revisions and corrections requested. Now your manuscript is accepted in PeerJ.